# The Antifungal Action Mode of *N*-Phenacyldibromobenzimidazoles

**DOI:** 10.3390/molecules26185463

**Published:** 2021-09-08

**Authors:** Monika Staniszewska, Łukasz Kuryk, Aleksander Gryciuk, Joanna Kawalec, Marta Rogalska, Joanna Baran, Anna Kowalkowska

**Affiliations:** 1Centre for Advanced Materials and Technologies CEZAMAT, Warsaw University of Technology, Poleczki 19, 02-822 Warsaw, Poland; jbaran@ch.pw.edu.pl; 2Department of Virology, National Institute of Public Health-National Institute of Hygiene, Chocimska 24, 00-791 Warsaw, Poland; lkuryk@pzh.gov.pl; 3Clinical Science, Targovax Oy, Lars Sonckin Kaari 14, Espoo Stella Luna Business Park, 02600 Espoo, Finland; 4Faculty of Chemistry, Warsaw University of Technology, Noakowskiego St 3, 00-664 Warsaw, Poland; alek.gryciuk@gmail.com (A.G.); joanna.kawalec.stud@pw.edu.pl (J.K.); martarogalska98@gazeta.pl (M.R.)

**Keywords:** *N*-phenacyldibromobenzimidazoles, *Candida* spp., *Cryptococcus neoformans*, action mode

## Abstract

Our study aimed to characterise the action mode of *N*-phenacyldibromobenzimidazoles against *C. albicans* and *C. neoformans*. Firstly, we selected the non-cytotoxic most active benzimidazoles based on the structure–activity relationships showing that the group of 5,6-dibromobenzimidazole derivatives are less active against *C. albicans* vs. 4,6-dibromobenzimidazole analogues (**5e**–**f** and **5h**). The substitution of chlorine atoms to the benzene ring of the *N*-phenacyl substituent extended the anti-*C. albicans* action (**5e** with 2,4-Cl_2_ or **5f** with 3,4-Cl_2_). The excellent results for *N*-phenacyldibromobenzimidazole **5h** against the *C. albicans* reference and clinical isolate showed IC_50_ = 8 µg/mL and %I = 100 ± 3, respectively. Compound **5h** was fungicidal against the *C. neoformans* isolate. Compound **5h** at 160–4 µg/mL caused irreversible damage of the fungal cell membrane and accidental cell death (ACD). We reported on chitinolytic activity of **5h**, in accordance with the patterns observed for the following substrates: 4-nitrophenyl-*N*-acetyl-β-d-glucosaminide and 4-nitrophenyl-β-d-*N*,*N*′,*N*″-triacetylchitothiose. Derivative **5h** at 16 µg/mL: (1) it affected cell wall by inducing β-d-glucanase, (2) it caused morphological distortions and (3) osmotic instability in the *C. albicans* biofilm-treated. Compound **5h** exerted *Candida*-dependent inhibition of virulence factors.

## 1. Introduction

Studies conducted during the past two decades have documented changes in the causative agents of nosocomial blood stream infections, and emphasized an increase of very critical fungal infections, particularly due to *Candida* spp. and *Cryptococcus* spp. [1]. The emergence of antifungal resistance required more concern to find out effective antimycotics with novel modes of action. Thus, introduction of *N*-phenacyldibromobenzimidazoles as another antimycotics destroying the fungal cell wall and membrane may be a milestone in the development of antifungal therapies. Moreover, treatment with anti-filamentation compound benefits the host by modulating immune responses [1]. An inhibition of morphological switch may provide an alternative approach to finding compounds with a potential to control the *Candida albicans* infections [2]. Morphogenesis is critical for biofilm formation, thus compounds able to inhibit sessile growth are needed [3].

Azoles are easy-to-use scaffolds in antifungal drug discovery [4]. Moreover, azoles are often functionalized with phenacyl group as a result of *N*-alkylation to gain excellent antifungal activity. There are known as biologically active *N*-phenacyl imidazoles [5,6,7,8,9,10], benzimidazoles [11,12,13,14], triazoles [5,7,15] or pyrazoles [16,17]. *N*-phenacyl azoles are often used as substrates for further synthesis of antifungal active agents [7,9,10,11,13,14,15,18,19,20]. In this study, we focused on dibromobenzimidazole synthesis due to promising antifungal activity rarely undertaken by scientists in worldwide studies on drug discovery, probably due to tedious synthesis [21,22,23,24]. 

We evaluated the toxicity of various *N*-phenacyldibromobenzimidazoles towards a mammalian cell line as well as the fungistatic and fungicidal effect against the *C. albicans* and *Cryptococcus neoformans* reference and clinical isolates resistant to azoles and echinocandins. The experiments with *N*-phenacyldibromobenzimidazoles have been encouraging in the current study because of the following action modes need to be assessed: 1. Phosphatidylserine externalization affecting subsequently the chitin content; 2. Cell wall stress induced by *N*-phenacyldibromobenzimidazoles resulted in the decreased/ increased ROS; 3. Lysosomotropic *N*-phenacyldibromobenzimidazoles exerted direct membrane lyses and caused osmotic pressure; 4. The concept regarding extensity of accidental cell death (ACD) under *N*-phenacyldibromobenzimidazoles. Since the primary targets of commercially available antimycotics are β-1.3-glucan and ergosterol, respectively, we underwent study if any compensatory mechanism in the cell wall and membrane occurs after the *N*-phenacyldibromobenzimidazole treatment. In our study, morphological changes enabled *N*-phenacyldibromobenzimidazoles gaining access to intracellular targets by facilitating membrane transience. 

## 2. Results

### 2.1. Synthesis of N-Phenacyldibromobenzimidazoles

As it is shown in Table 1 and Figure 1, compounds **4**–**5** were synthesized by *N*-alkylation of 5,6-dibromobenzimidazole **1** or 4,6-dibromobenzimidazole **2** with phenacyl chlorides or bromides **3** in the presence of K_2_CO_3_ in MeCN. The time of the reactions as well as the yields depended on the structure of the phenacyl derivative **3** used. In the case of unsubstituted phenacyl bromide **3a** and monofunctionalized derivatives **3b**–**d**, the respective products **4a**–**d** and **5a**–**d** were isolated in 72–94%. Meanwhile, in reactions with alkylating agents, **3e**–**j** possessing two or three halogen atoms in the benzene ring, afforded complicated mixture of products, so the target compound **4e**–**j** and **5e**–**j** were isolated in 13–24% [25]. All *N*-phenacylbenzimidazoles **4**–**5** were purified by column chromatography, followed by crystallization.

### 2.2. The Antifungal Effect of Dibromobenzimidazole Derivatives

As it is shown in Table 2 and Appendix A, in our initial screening of twenty dibromobenzimidazole derivatives we assessed the percentage of cell growth inhibition (%I). At the inhibitory concentration of 50% (IC_50_), the concentration of benzimidazoles that reduces the cell growth of *C. albicans* SC5314 by ≥50% was determined. Secondly, randomly selected (**5f**) and the most effective inhibitors (**5e** and **5h**) were tested against the *C. albicans* SPZ176 isolate resistant to Flu and Itr (Table 2). Further, **5e** displayed IC_50_ at 4–16 µg/mL (Table 2) and the mode of fungicidal action against SC5314 at 8–16 µg/mL (lg R ≤ 1.19 in Table 3). **5f** showed lg R = 1 at 8 µg/mL (Table 3). Contrariwise, **5h** displayed no candidacidal action (lg R ≤ 0.43 in Table 3). Moreover, a paradoxical growth phenomenon of the reference strain SC5314 [26] was noted for the following derivatives: **4f**, **4h**, **5a,** and **5g**–**i** (Appendix A) as well as **5b**, **5e**–**f**, **5h**, **5j** (Table 2). Briefly, we noted a slow decrease in the viable cell growth at higher concentrations (e.g., %I = 53 ± 8 at 16 µg/mL for **5b**) vs. the lowest concentrations at which the cell growth was substantially inhibited (e.g., %I = 95 ± 8 at 8 µg/mL for **5b**).

We determined the effectiveness of dibromobenzimidazole derivatives against the fungal isolates using colony forming unites (cfu) assay (Table 4). The exhaustive data clearly demonstrated that cfu were recovered after treatment with the tested dibromobenzimidazoles (Table 4). The most effective **5h** at 16 µg/mL totally inhibited recovery of cfu of both clinical isolates. In the case of *C. neoformans*, there was no cfu recovery after treatment with **5h** at the concentration range of 8–16 µg/mL. Thus, *C. neoformans* was more sensitive to **5h** than *C. albicans*. We identified the leading fungicidal compound **5h** to be used in a series of follow-up analyses to establish its action mode in vitro.

### 2.3. Cytotoxicity of N-Phenacyldibromobenzimidazole Derivatives

As it was shown in Figure 1, the Vero cell viability or cytotoxicity generated by the most active compounds (fungicidal) was assessed using the MTS method. Figure 1 indicates CC_50_ = 32–64 µg/mL and CC_90_ = 64–256 µg/mL for **5e** and **5f**. Moreover, **5h** displayed CC_50_ = 32–64 µg/mL and CC_90_ = 256 µg/mL. Thus, all compounds did not inhibit the NAD(P)H dehydrogenase (quinone) activity and disturb cell membrane permeability.

### 2.4. Antifungal Activity of ***5h*** in Combination with Osmoprotectant

The %I values of **5h** were changed in the presence of sorbitol, and it suggests influence of **5h** on the cell wall structure of the *C. albicans* clinical isolate (Table 5). In details, **5h** displayed lack or weak (%I = 8 ± 18) cell growth inhibition at 16 µg/mL in the presence of 0.8 M sorbitol as an osmotic protectant in the medium vs. one without sorbitol (%I = 100 ± 3 in Table 2). For the *C. neoformans* isolate, the antifungal activity of **5h** was as follows: (1) **5h** at 4 µg/mL causes no cell growth inhibition in medium with sorbitol added vs. 1 × 10^5^ cfu/mL recovered in medium without sorbitol (Table 4); (2) **5h** displays no growth recovery at 8–16 µg/mL in medium without sorbitol vs. %I = 79–95 at the same range of concentrations in medium with sorbitol added.

### 2.5. Chitinolytic Activity of ***5h***

As it was shown in Table 6, the detailed studies on chitinolytic activity showed affinity of **5h** to the following substrates: 4-nitrophenyl-*N*-acetyl-β-d-glucosaminide and 4-nitrophenyl-*N*,*N*′-diacetyl-β-d-chitobioside. Contrariwise, **5h** displayed no affinity to 4-nitrophenyl-β-d-*N*,*N*′,*N*″-triacetylchitothiose.

### 2.6. Efflux Disorder under ***5h***

Rho123 was not able to leave the mitochondrion due to the membrane potential decreased (efflux decreased) as a results of cell death. For the *C. albicans* ref. strain and *C. neoformans* isolate, efflux decreased in line with increased conc. of **5h** (Table 7). Contrariwise, in the case of *C. albicans* clinical isolate, efflux was noted for the **5h**-treated cells at 16 μg/mL (Table 7).

### 2.7. Compound ***5h*** Induces ROS Generation

Treatment of the fungal cells with low concentration of **5h** led to the ROS production at high level (198%In Figure 2). Generally, in the case of *C. neoformans*, the level of ROS production increased in line with decreasing concentrations of **5h**. Remaining strains showed ROS under detectable level, with exception of *C. albicans* SPZ176 generating ROS at 22% under treatment with **5h** at 4 µg/mL.

### 2.8. Estimation of Accidental Cell Death in the ***5h***-Treated Fungi

As shown in Figure 3 and Figure 4, **5h** at the concentrations ranging from 4 to 160 µg/mL generated necrosis (accidental cell death ACD) in the fungal cells and protoplasts. Apoptosis early or late was induced approx. at 0.31% or 0.83% in the *C. albicans* protoplasts under treatment with **5h** at 160 µg/mL. In the case of *C. neoformans*, apoptosis was generated approx. at 0.04% (early) and 0.02% (late) by **5h** at 160 µg/mL. In the case of the C*. neoformans* protoplasts, late apoptosis was noted approx. at 0.03% or 0.02%, respectively, for 160 or 16 µg/mL.

### 2.9. Antifungal Action and Accidental Cell Death by Fluorescent Structural Staining Techniques

The resulting cell wall damage and cell viability were assessed using Confocal laser scanning microscopy (CLSM) after treatment with **5h** (twelve images were assessed for each treatment/staining). As it was shown using CFW staining (Figure 5), **5h** at 16 µg/mL induced the cell wall rearrangement of the *C. albicans* sessile conglomerate. Biofilm’s chitin content was redistributed and elevated under treatment with **5h** (vivid blue fluorescence of elevated chitin in Figure 5). Contrariwise, action of **5h** against the *C. neoformans* sessile growth was not significant (Figure 6). In Figure 6, very few cells were totally stained with CFW in conglomerate vs. the untreated control showing several cells with bright blue fluorescence. Thus **5h** did not reorganize the cell wall chitin content of *C. neoformans*.

Congo red (CR) interacts with β-d-glucan of the **5h**-treated *C. albicans* sessile cells (Figure 7). Thus, the cells exposed to **5h** at 16 µg/mL exhibit increased frequencies of the cell wall damage (arrows in Figure 7). Contrariwise, the biofilm of ***C****. neoformans* treated with **5h** was found CR sensitive in comparable level to the untreated sessile cells (Figure 8). Thus **5h** did not disturb the glucan content of *C. neoformans*.

Compound **5h** altered plasma membrane permeability, which is indicated by intensive red fluorescence of the **5h**-treated sessile cells (Figure 9), compounds induced necrosis-like cell death (bright red fluorescence of ethidium bromide EB inside the damaged sessile cells in Figure 9). Contrariwise, *C. neoformans* was resistant to **5h** (arrows indicate weak green fluorescence of acridine orange AO inside the viable cells in Figure 10). 

## 3. Discussion 

Antifungal structure–activity relationships showed that the group of 5,6-dibromobenzimidazole derivatives are less active against *C. albicans* vs. 4,6-dibromobenzimidazole analogues (Table 2 and Appendix A). Moreover, the substitution of chlorine atoms to the benzene ring of the *N*-phenacyl substituent extended anti-*C. albicans* activity (**5e** with 2,4-Cl_2_ or **5f** with 3,4-Cl_2_ in Table 1, Table 2, Table 3 and Table 4 and Appendix A). Contrariwise, the substitution of bromine or fluorine atoms in the same positions influences weak activity against *Candida* spp. The findings described above are in line with Vargas-Oviedo et al. [14]. It is worth to mention that **5h** substituted with fluorine atoms at C2 and C4 of the benzene ring of the *N*-phenacyl group exhibited excellent fungicidal activity against the *C. albicans* reference and clinical strain as well as the *C. neoformans* isolate (Table 2, Table 3 and Table 4). In our study, the leading compound **5h** (Table 4) was <16-times less active than AmB with minimal fungicidal concentration MFC_90_ = 1 µg/mL [31] and MFC = 0.5 [32] against the *C. albicans* isolates and SC5314, respectively. Structure–activity relationships provide opportunities for synthesis of dibromobenzimidazole analogues with improved antifungal action. Moreover, the most active antifungals (**5e**–**f**, **5h**) at the concentration range of 32–0.125 µg/mL were developed to generate viable and vital eukaryotic cells (Figure 1 and Appendix A; Appendix A). Thus, the tested dibromobenzimidazole were proved to be less cytotoxic against the Vero cells compared to AmB (toxic at 15–20 µg/mL after 24 h) [33].

In line with the results obtained in the presence of the osmo-protectant in the growth medium [34], we showed that **5h** is the *C. albicans* cell wall inhibitor, displaying reverse effect in the presence of sorbitol (Table 5). The effect is characterized by decreasing in %I (Table 5) as observed in the medium with sorbitol vs. medium without protectant (Table 2). Our studies demonstrated that osmotic protector reduces anti-*Candida* activity of **5h**. In alignment with Górska-Nieć et al. [35], we proved that enhanced biomass production leads to loss of antifungal activity of **5h** at concentrations ranging from 4 to 16 µg/mL. Moreover, the activity of **5h** did not correspond with AmB affecting cell wall due to activity accompanied by an increase concentration in medium with sorbitol [36].

Moreover, the micromorphological evaluation of the *C. albicans*-treated with **5h** revealed the lack of structures indicating fungal mycelium typical for biofilm. Thus **5h** was able to inhibit the biofilm formation (Figure 5, Figure 6, Figure 7, Figure 8, Figure 9 and Figure 10). Since the yeast-hyphae morphological transition is relevant for *C. albicans* virulence [37] we indicated that **5h** represents promising therapeutic. 

We showed that **5h** acts by the cell wall chitin lysis originated from the comparison studies with chitinase (Table 6). Chitin is a polymer of β-1,4-linked *N*-acetyl-d-glucosamine (GlcNAc), which is an integral component of the fungal cell wall [38]. **5h** was able to hydrolase 4-nitrophenyl-*N*-acetyl-β-d-glucosaminide and 4-nitrophenyl-β-d-*N*,*N*′,*N*″-triacetylchitothiose without activity against triacetylchitothiose (Table 6). Based on our results and in line with Nielsen and Sörensen [39], we hypothesized that **5h** displays comparable activity to chitinase (EC 3.2.1.14). Since the ability of Congo red CR fluorescent tracker to visualize the fungal cell wall elements was described previously [40,41], we used CR as a diazo compound pertaining to its high affinity to polysaccharides in the **5h**-treated *C. albicans* [40,41]. In line with Shalmy et al. [40] we found good staining result of CR in the **5h**-treated *C. albicans* vs. **5h**-treated capsule of *C. neoformans* which was poorly stained (Figure 7 vs. Figure 8). We showed that **5h** displayed *Candida* spp. dependent activity.

Furthermore, **5h** induced the phosphatidylserine PE translocation and membrane permeability [41], these were shown using the Annexin V and propidium iodide PI staining assay (Figure 3 and Figure 4). We hypothesized that PE externalization affect subsequently the elevated chitin content (Figure 5) and activity of **5h**. Moreover, this polymer play an essential role in the sensitivity (or resistance) of *C. albicans* to AmB [42,43,44]. Since ROS play a crucial role in intracellular signalling [44], *C. neoformans* treated with **5h** displayed elevated ROS (Figure 2 and Appendix A) regarded as a cell death phenotype in connection with plasma membrane disintegration (at 16–160 µg/mL in Figure 3 and Figure 4 and Appendix A) and loss of clonogenicity (at 8–16 µg/mL in Table 4). In details, the high levels of ROS at the **5h**-treated cells at 4 µg/mL activate apoptosis pathway capable of inducing ACD (Figure 2). We hypothesized that the adaptive response of *C. neoformans* showing elevated ROS production promotes stress resistance to **5h**. Contrariwise, decrease in the ROS level by incubation with **5h** can induce the lethal process adequately monitored by cytometric analysis (Figure 3 and Figure 4 and Appendix A). Based on the latter findings, **5h** can act such as anti-oxidant. Moreover, elevated ROS under **5h** correlated with fungicidal effect typical for AmB [44].

We used Rho123 as a membrane-potential-sensitive cationic fluorophore [45] to show that it was not able to leave the mitochondrion due to decreased membrane potential as a result of the **5h**-treated cell death (Figure 2, Figure 3 and Figure 4, Figure 9 and Figure 10 as well as Appendix A). We concluded that **5h** can be mitochondrial inhibitor of *C. albicans* ref and *C. neoformans*. Contrariwise, the Rho123 efflux simply increased pump activity in the **5h**-treated *C. albicans* isolate resistant to azoles. The compelling evidence for reduced filamentation and ACD (progenitors of mycoses) are targets for dibromobenzimidazole. Finally, our findings suggested a general strategy for antimycotics development that might be useful in limiting the emergence of fungal resistance. We selected **5h** as the most compound with significant response against the fungal virulence factors. We propose that **5h** acts synergistically to disrupt the *C. albicans* cell wall/membrane. These structures establish an excellent target for specific inhibition of pathogenic fungi.

## 4. Materials and Methods

### 4.1. General Remarks of the N-Phenacyl Dibromobenzimidazole Derivatives Synthesis

Commercially available reagents from Sigma Aldrich (Darmstadt, Germany), Fluka (Charlotte, NC, USA) and Avantor (Gliwice, Poland) were used as supplied. The measured melting points were not corrected. The column chromatography was performed using Silica gel 60 (Merck) of 40–63 μm. Thin-layer chromatography was carried out on TLC aluminium plates with silica gel Kieselgel 60 F_254_ (Merck, Darmstadt, Germany) (0.2 mm thickness film). The ^1^H and ^13^C NMR spectra were measured with a Varian 500 spectrometer operating at 500 MHz for ^1^H and 125 MHz for ^13^C nuclei. Chemical shifts (δ) are given in parts per million (ppm); signal multiplicity assignment: s, singlet; d, doublet; dd, doublet of doublets; m, multiplet; coupling constant (J) are given in hertz (Hz). High resolution mass spectrometry (HRMS) was carried out on Q Exactive Hybrid Quadrupole-Orbitrap Mass Spectrometer (Bremen, Germany), ESI (electrospray) with spray voltage 4.00 kV at Institute of Biochemistry and Biophysics Polish Academy of Science (IBB PAS, Warsaw, Poland. The most intensive signals are reported.

#### 4.1.1. Synthesis of **4a**–**d** and **5a**–**d**

To a stirred suspension of 5,6-dibromobenzimidazole **1** or 4,6-dibromobenzimidazole **2** (1 mmol, 0.276 g) in MeCN (20 mL) K_2_CO_3_ (4 mmol, 0.553 g) followed by **3a**–**d** (1 mmol) was added. The reaction was carried out at room temperature (20–22 °C) for 24 h. After this time the solid products were filtered, washed out with MeCN (25 mL), evaporated. The residue was purified by column chromatography (silica gel/CHCl_3_, eluent CHCl_3_). Analytical sample was crystallized (EtOH).

#### 4.1.2. Synthesis of **4e**–**i** and **5e**–**i**

To a stirred suspension of 5,6-dibromobenzimidazole **1** or 4,6-dibromobenzimidazole **2** (1 mmol, 0.276 g) in MeCN (20 mL) K_2_CO_3_ (8 mmol, 1.106 g) followed by **3e**–**j** (2 mmol) was added. The reaction was carried out at room temperature (20–22 °C) for 3h for **3e**-**i** and 96 h for **3j**. After this time the solid products were filtered, washed out with MeCN (25 mL), evaporated. The residue was purified twice by column chromatography (silica gel/CHCl_3_, eluent CHCl_3_ followed by silica gel/toluene, eluent toluene/EtOAc gradient, 50:0 to 50:15). Analytical sample was crystallized (EtOH).

### 4.2. Biological Studies

#### 4.2.1. Yeast Cultures

Antifungal activity of new *N*-phenacyldibromobenzimidazole derivatives was carried out against two *C. albicans* strains: reference *C. albicans* SC5314 from American Type Culture Collection (ATCC) and clinical SPZ176 strain (resistant to antifungal drugs: fluconazole Flu and itraconazole Itr) and clinical *C. neoformans* SPZ173 strain (naturally resistant to echinocandins). Fungal strains were stored at −80 °C in Microbank system (ProLab Diagnostics, Richmond Hill, ON, Canada) and cultured for 24 h at 30 °C with shaking at 100 rpm prior to each examination in liquid medium: YEPD (Yeast Extract Peptone Dextrose) or YNB (Yeast Nitrogen Base 0.67% *w*/*v*, glucose 2% *w*/*v*, CSM-URA 0.077% *w*/*v*, sterile water). After centrifugation at 3000 rpm at 4 °C for 5 min, cells were washed twice with sterile water and resuspended to prepare suspensions for experiments (ranging from 1.9 × 10^7^ to 2.0 × 10^11^ cfu/mL; where cfu/mL = (number of colonies) × (inverse dilution of coefficient plated) × 10.

#### 4.2.2. Broth Microdilution Assay: MIC and MFC Determination

Stock solutions of 1600 µg/mL were prepared by dissolving the following compounds: **4a**, **4j**, **5b**, **5d**, **5e**, **5f**, **5h**, and **5j** in 96% DMSO. Concentrations of 800, 400 and 200 µg/mL were later prepared form stock solutions and stored at −20 °C. Antifungal susceptibility testing was performed by broth microdilution assay according to the method M27-A3 by CLSI (Clinical and Laboratory Standards Institute) [27]. The microtiter plates were prepared containing compound test wells (CTW), sterility control wells (STW) and growth control wells (GCW) in triplicate in YEPD or YNB liquid medium. Compounds were added to proper wells (CTW and STW) to final concentration of 16, 8 and 4 µg/mL. Initial yeast suspensions (prepared as described above) were diluted 105-fold in sterile water and 20-fold in liquid medium before examination and then added to wells (CTW and GCW). To obtain the same concentration of DMSO in each well, DMSO was also added to growth control wells. Microtiter plates were incubated for 48 h at 30 °C. After 48 h visual assessments and absorbance measurements at 405 nm were performed using Synergy H4 Hybrid Reader (BioTek Instruments, Winooski, VT, USA). Antifungal activity was calculated as the percentage of cell growth inhibition using formula: % of inhibition = 100 × (1 − (ODCTW − ODSCW)/(ODGCW − ODSCW), were OD means absorbance of each well. CTWs containing each concentration of tested compounds were mixed and diluted 10^4^-fold in sterile water. Then, 100 µL of each suspension was spread on the plates containing solid YEPD or YNB medium and incubated at 30 °C for 48 h. After 48 h, visual assessments were performed and Colony Forming Unit per 1 mL (cfu/mL) was calculated. Logarithmic cfu growth reduction factor (R) was calculated by formula: R = log (cfu/mL GCW) − log (cfu/mL CTW). Minimum Fungicidal Concentration (MFC) was determined as the concentration which resulted in ≥99.9% CFU/mL reduction (R > 3).

#### 4.2.3. Determination of **5e**–**f** and **5h** Cytotoxicity

Cytotoxicity evaluation was performed using MTS reagent (3-(4,5-dimethylthiazol-2-yl)-5-(3-carboxymethoxyphenyl)-2-(4-sulfophenyl)-2*H*-tetrazolium, MTS, Promega, USA) against mammal Vero cell line (ATCC CCL-81, LGC Standards, Lomianki, Poland). Vero cell line was cultured in vitro at 37 °C and 5% CO_2_ in EMEM medium (Eagle’s Minimum Essential Medium, Sigma-Aldrich, St. Louis, MO, USA) supplemented with 10% FBS (foetal bovine serum, Gibco, Waltham, MA, USA) and 1% antibiotics. Cells were passaged several times and eventually transferred to microtiter plate (final density of 400,000 cells per mL) and incubated for 24 h prior to examination [46]. Resulting cell monolayer was maintained in EMEM medium supplemented with 10% FBS. Stock solutions of each comp. were prepared (conc. of 512 μg/mL) and added in triplicate in 2-fold dilutions to the plate until final conc. of 0.125 μg/mL. Positive control with cells and without tested comp. and negative control without cells were also prepared. After 24 h of incubation, 10 μL of MTS reagent was added to each well and the plates were incubated for 3 h in darkness [46]. Finally, the absorbance at 490 and 660 nm was measured with Synergy H4 Hybrid Reader (BioTek Instruments, Winooski, VT, USA) and specific absorbance (SA) was calculated as follows: SA = A_490_ − A_660_. Viability of Vero cells was calculated using formula: % viability = (SA Test − SA Blank) / (SA Positive control − SA Blank) × 100, and the cytotoxicity of the compounds: % cytotoxicity = (SA Positive control − SA Test)/(SA Positive control − SA Blank) × 100 [41].

#### 4.2.4. Broth Microdilution Assay: Activity of **5h** Accompanied by Osmotic Protector

The evaluation of antifungal activity of **5h** against clinical *C. albicans* SPZ176 and *C. neoformans* SPZ173 strains was performed by the CLSI M27-A3 method described above with modifications. Compound test wells (CTW), sterility control wells (STW) and growth control wells (GCW) were prepared as previously mentioned in liquid medium consisting of YNB and 0.8 M sorbitol (Sigma-Aldrich, USA) as an osmotic protector [34]. Plates were incubated for 120 h at 30 °C. Absorbance was measured at 405 nm after 96 and 120 h of incubation using Synergy H4 Hybrid Reader (BioTek Instruments, USA). Antifungal activity was calculated as the percentage of cell growth inhibition using formula presented above.

#### 4.2.5. Examination of Chitinolytic Activity of **5h**

Test was preformed using Chitinase Assay Kit (CS0980, Sigma-Aldrich, USA). Procedure was based on technical bulletin obtained from producer [30]. Four groups of samples were prepared on microtiter plate: (1) Blanc—40 μL of substrate A, B or C; (2) Standard—120 μL of standard solution (included in Assay Kit); (3) Test—36 μL of substrate A, B or C with **5h** to final concentration of 16 μg/mL (4 μL of **5h** at 160 mg/mL); (4) Control—36 μL of substrate A, B or C with 4 μL of 0.2 mg/mL chitinase. Plate was incubated for 30 min at 37 °C and then the reaction was stopped with stop solution form the Assay Kit. Absorbance at 405 nm was measured using Synergy H4 Hybrid Reader (BioTek Instruments, USA) [27]. Chitinolytic activity was calculated using formula:(1)$$\mathrm{ACT}=\frac{\left(A_{T}-A_{B}\right)\;\times 0.05\;\times \;0.07\;\times \;DF}{A_{S}\;\times t\;\times \;V_{\mathrm{PR}}}$$
where: ACT—chitinolytic activity [U/mL]; $A_{T}$—absorbance of test sample at 405 nm [-]; $A_{B}$—absorbance of blank at 405 nm [-]; 0.05—*p*-nitrofenol concentration in standard solution [μmol/mL]; 0.07—final volume of samples in each test well (after addition of stop solution) [mL]; DF—enzyme dilution factor (here equal to 1—enzyme was not diluted); $A_{S}$—absorbance of standard [-]; t—reaction duration time [min]; $V_{\mathrm{PR}}$—volume of **5h** or chitinase [mL] (here 0.004 mL).

#### 4.2.6. Determination of the Rhodamine 123 Efflux from the Cells Treated with **5h**

*C. albicans* SC5314 ref. strain’s, *C. albicans* SPZ176 clinical strains and *C. neoformans* SPZ173 clinical strain’s culture were prepared as previously described. Test samples were prepared by adding of 100 μL of 10^5^-fold diluted cells suspensions to 900 μL YNB medium with **5h** at conc. of 160, 16 or 4 μg/mL. Control was obtained by adding 10^5^-fold diluted cells suspensions to 900 μL YNB without **5h**. All samples were incubated at 30 °C with shaking at 120 rpm for 18 h. Suspensions were then centrifuged at 9500 rpm for 2 min and cells were washed with PBS. Following, 100 μL of the washed cells were added to 900 μL of PBS with glucose (5 mM) and rhodamine B (7.18 mg/mL) (Sigma-Aldrich, Darmstadt, Germany). After 30 min of incubation at 37 °C, suspensions were centrifuged at 9500 rpm for 2 min and the cells were washed with PBS. Then, the cells were resuspended in PBS with glucose (1 mM) and incubated at 37 °C with shaking at 120 rpm for 18 h. Then, the post growth medium was separated from the cells by centrifugation at 9500 rpm for 2 min and 20 μL of supernatant was added to microtiter plate. To prepare 10-fold diluted samples, 180 μL of sterile water was added to each well. Fluorescence was measured with excitation at 521 nm and emission at 627 nm using Synergy H4 Hybrid Reader (BioTek Instruments, Winooski, VT USA). Concentration of Rho123 was calculated using formula: C = (E − Blank − 1151.2) × 10/556.91; where: E—emission; Blank—emission of PBS/glucose medium; 1151.2 and 556,91—coefficients of rhodamine standard curve; 10—dilution coefficient. Decrease of the Rho123 content was determined using the formula: ΔC% = [C(Test) − C(Control)]/C(Control) × 100; where: C(Test)—concentration of rhodamine in tested samples; C(Control)—concentration of rhodamine in control samples [45].

#### 4.2.7. Determination of Reactive Oxygen Species (ROS) Concentration after Incubation with **5h**

Examination was preformed using DCFDA/H2DCFDA kit (Thermo Fisher Scientific, Waltham, MA, USA) [45]. *C. albicans* SC5314, *C. albicans* SPZ 176 clinical isolate and *C. neoformans* SPZ 173 clinical isolate were prepared as previously described. Test samples were prepared as described in Rho123 assay (see Section 4.2.6). Positive control treated with hydrogen peroxide at conc. of 3% and untreated negative control were used. Test and control tubes were incubated for at 30 °C with shaking at 120 rpm for 18 h. Suspensions were then centrifuged at 5000 rpm for 5 min and cells were washed with PBS. Then, 999.5 μL of the cell suspension was transferred to new test tube and 0.5 μL of fluorescein solution (10 mM) in DMSO (96%) was added. All samples were incubated for at 30 °C with shaking at 120 rpm for 40 min. Test samples (without positive control) were centrifuged at 5000 rpm for 5 min and cells were resuspended in YNB medium. Then, all samples were incubated at 30 °C with shaking at 120 rpm for 18 h. The positive control was transferred on a microtiter plate as well as 10-fold dilution of test samples. Fluorescence was measured with extinction at 485 nm and emission at 530 nm using Synergy H4 Hybrid Reader (BioTek Instruments, Winooski, VT, USA). Change in ROS concentration was calculated using formula: ΔC = [E(Test) − E(Control)] × 100%/E(Control); where ΔC—change in ROS concentration; E(Test)—fluorescence of test samples; E(Control)—fluorescence of negative control [45].

#### 4.2.8. Cytometric Analysis of Cell Death Type

To determine the type of cell death induced by the action of **5h**, flow cytometry analysis was performed using the protoplasts and *C. albicans* SPZ176 and *C. neoformans* SPZ173 cells. Protoplasts were obtained according to the method previously described [41]. Cells and protoplasts were then incubated with 160, 16 or 4 μg/mL of **5h** at 30 °C with shaking at 120 rpm for 24 h. Compound-free growth controls were also prepared. After harvesting by centrifugation at 3000 rpm at 4 °C for 5 min; cells were washed and resuspended with sterile water. Determination of the cell death type was conducted by staining using annexin V and propidium iodide (FITC Annexin V/Dead Cell Apoptosis Kit with FITC annexin V and PI, for Flow Cytometry, (Invitrogen, Waltham, MA, USA) [42]. Suspensions were diluted by 10-fold with the proper buffer from the kit and then incubated for 10–15 min with 1 μL of annexin. After centrifugation at 3000 rpm at 4 °C for 5 min cells and protoplasts were resuspended in the buffer and incubated in ice for 5–15 min with 1 μL of propidium iodide (PI). Fluorescence was analysed by flow cytometry using BD FACSLyrics 2L6C with FACSuite Software 1.4 RUO (BD Biosciences, Mississauga, ON, Canada).

#### 4.2.9. Confocal Laser Scanning Microscopy (CLSM) Analyses of the *C. albicans* and *C. neoformans* Biofilms Treated with **5h**

*C. albicans* SPZ 176 and clinical *C. neoformans* SPZ 173 cultures were prepared, as previously described [41]. Suspensions were centrifuged at 5000 rpm at 4 °C for 5 min and resuspended in 2 mL of the YNB medium. Then, 500 μL of suspensions were placed on coverslips on the bottom of a 24 well plate (two wells were prepared for each strain). Plate was then incubated at 37 °C for 24 h without shaking. Then, the plate was washed twice with PBS. To the tested well, 500 μL of **5h** solution in PBS (final conc. of 16 μg/mL) was added (test sample), to the control one 500 μL PBS was added. Plate was incubated at 37 °C for 18 h. Biofilms were then washed twice with PBS and then 495 μL of PBS and 5 μL of staining solution was added. The following staining solutions were used: CR (Congored, Sigma-Aldrich) at stock conc. of 200 µg/mL; CFW (Calcofluor White, Sigma-Aldrich) at stock conc. of 250 µg/mL; AO (Acridine Orange, Roche Diagnostics GmbH, Mannheim, Germany) at stock conc. of 100 µg/mL; EB (Ethidium Bromide, Roche Diagnostics GmbH) at stock conc. of 100 µg/mL [41]. Final staining solutions were diluted by 100-fold. The plate was incubated at 37 °C for 18 h. Microscope observations were carried out using confocal laser scanning microscopy (CLSM) with Olympus FLUOREVIEW FV1000 (Olympus, Osaka, Japan).

## Data Availability

Not available.

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
