# Peer review of "The Antifungal Action Mode of N-Phenacyldibromobenzimidazoles"

_molecules, 2021, doi:10.3390/molecules26185463_

Round 1

Reviewer 1 Report

The manuscript “The Antifungal Action Mode of N-phenacyldibromobenzimidazoles” has as main goal the characterisation of the action mode of N-phenacyldibromobenzimidazoles against C. albicans and C. neoformans. It is an important work as the need for active antifungals are urgent due to the resistance of many microorganism to the current antimycotics used.

However, there are some issues that need to be enlightened

The denomination of microorganisms should be in italic (ie C. albicans), in the text there are many of them that should be corrected, a carefully revision of the text should be made.

Table 2 is not very clear in its information content. Although authors speak in MICs, they are showing percentages of reduction. The concentration units must be presented in the table. Also, table 3 is very confusing. The information explained on paragraph 2.2 it is not in accordance to what is shown in both tables. It is not clear how a higher concentration result in a lower percentage of inhibition. It is not well stated what is the importance of data in table 4. All the 2.2. section must be revised.

Section 2.4. should be: “2.4. 5h’s activity…”.

It is not very well explained why from section 2.4 only compound 5h is tested.

In section 2.8. six figures (figure 5 to figure 10) are presented with no explanation about them! Only two are mentioned in the text (and they are wrong, it should be figure 5 and 6, not 4 and 5 as it is stated). Reference to these figures must be done in text. Readers need to understand what is intended to be shown with these pictures.

Also the fact that the work is carried out with only 3 strains leaves some doubts. More strains should be used.

Author Response

Reviewer #1

Authors’ answers

The denomination of microorganisms should be in italic (ie C. albicans), in the text there are many of them that should be corrected, a carefully revision of the text should be made.

Italic was corrected

Table 2 is not very clear in its information content. Although authors speak in MICs, they are showing percentages of reduction. The concentration units must be presented in the table.

InTable 2, MIC was removed

(% I) means the cell growth inhibition, it was presented in Table 2

Conc. units were presented in Table 2

Table 2 was rewritten

Also, table 3 is very confusing.

Table 3 was rewritten and extensive legend was included

Additionally Table 5 was rewritten

The information explained on paragraph 2.2 it is not in accordance to what is shown in both tables. It is not clear how a higher concentration result in a lower percentage of inhibition.

It was explained, please see below:

Moreover, a paradoxical growth phenomenon [26] was noted for the following derivatives: 4f, 4h, 5a (Fig. S5, Fig. S7, Fig. S9, Fig. S11-S12) as well as 5b, 5e-f, 5h, 5j (Table 2). Briefly, we noted a slow decrease in the viable cell growth at higher concentrations (e.g., % I=53±8 at 16 µg/ml for 5b) vs the lowest concentrations at which the cell growth was substantially inhibited (e.g., % I=95±8 at 8 µg/ml for 5b).

It is not well stated what is the importance of data in table 4.

The importance of Table 4 including the cfu assay’s results was explained, see below:

We determined the effectiveness of dibromobenzimidazole derivatives against the fungal isolates using colony forming unites (cfu) assay (Table 4). The exhaustive data clearly demonstrated that cfu were recovered after treatment with the tested dibromobenzimidazoles (Table 4). The most effective 5h at 16 µg/ml totally inhibited recovery of cfu of both clinical isolates. In the case of C. neoformans, there was no cfu recovery after treatment with 5h at the concentration range of 8-16 µg/ml. Thus, C. neoformans was more sensitive to 5h than C. albicans. We identified the leading fungicidal compound 5h to be used in a series of follow-up analyzes to establish its action mode in vitro.

All the 2.2. section must be revised.

The section 2.2 was totally revised, see below:

As it was shown in Table 2 and Fig. S1 -S12 (in Supplementary file), in our initial screening of twenty dibromobenzimidazole derivatives we assessed the percentage of cell growth inhibition (%I). The inhibitory concentration of 50% (IC50), the concentration of benzimidazoles that reduces the cell growth of Candida albicans SC5314 by ≥50% was determined. Secondly, randomly selected (5f) and the most effective inhibitors (5e and 5h) were tested against the C. albicans SPZ176 isolate resistant to Flu and Itr (Table 2). 5e displayed IC50 at 4-16 µg/ml (Table 2) and the mode of fungicidal action against SC5314 at 8-16 µg/ml (lg R≤1.19 in Table 3). 5f showed lg R=1 at 8 µg/ml (Table 3). Contrariwise, 5h displayed no candidacidal action (lg R≤0.43 in Table 3). Moreover, a paradoxical growth phenomenon of the reference strain SC5314 [26] was noted for the following derivatives: 4f, 4h, 5a (Fig. S5, Fig. S7, Fig. S9, Fig. S11-S12) as well as 5b, 5e-f, 5h, 5j (Table 2). Briefly, we noted a slow decrease in the viable cell growth at higher concentrations (e.g., % I=53±8 at 16 µg/ml for 5b) vs the lowest concentrations at which the cell growth was substantially inhibited (e.g., % I=95±8 at 8 µg/ml for 5b).

We determined the effectiveness of dibromobenzimidazole derivatives against the fungal isolates using colony forming unites (cfu) assay (Table 4). The exhaustive data clearly demonstrated that cfu were recovered after treatment with the tested dibromobenzimidazoles (Table 4). The most effective 5h at 16 µg/ml totally inhibited recovery of cfu of both clinical isolates. In the case of C. neoformans, there was no cfu recovery after treatment with 5h at the concentration range of 8-16 µg/ml. Thus, C. neoformans was more sensitive to 5h than C. albicans. We identified the leading fungicidal compound 5h to be used in a series of follow-up analyzes to establish its action mode in vitro.

Tables: 2-4 were revised, please see the revised manuscript. The legends of these Tables were rewritten (please, see in the revised manuscript)

Section 2.4. should be: “2.4. 5h’s activity…”.

It was corrected, see below:

2.4. Antifungal activity of 5h…

It is not very well explained why from section 2.4 only compound 5h is tested.

It was explained, see below:

We identified the leading fungicidal compound 5h to be used in a series of follow-up analyzes to establish its action mode in vitro.

In section 2.8. six figures (figure 5 to figure 10) are presented with no explanation about them! Only two are mentioned in the text (and they are wrong, it should be figure 5 and 6, not 4 and 5 as it is stated).

In the section 2.8 all figures were explained. Figures were cited correctly, please see below:

As it was shown using CFW staining (Fig. 5), 5h at 16 µg/ml induced the cell wall rearrangement of the C. albicans sessile conglomerate. Biofilm’s chitin content was redistributed and elevated under treatment with 5h (vivid blue fluorescence of elevated chitin in Fig. 5). Contrariwise, action of 5h against the C. neoformans sessile growth was not significant (Fig. 6). In Fig. 6, very few cells were totally stained with CFW in conglomerate vs the untreated control showing several cells with bright blue fluorescence. Thus 5h did not reorganize the cell wall chitin content of C. neoformans.

Congo red (CR) interacts with β-D-glucan of the 5h-treated C. albicans sessile cells (Fig. 7). Thus the cells exposed to 5h at 16 µg/ml exhibit increased frequencies of the cell wall damage (arrows in Fig. 7). Contrariwise, the biofilm of C. neoformans treated with 5h was found CR sensitive in comparable level to the untreated sessile cells (Fig. 8). Thus 5h did not disturb glucan content of C. neoformans.

Compound 5h altered plasma membrane permeability which is indicated by intensive red fluorescence of the 5h-treated sessile cells (Fig. 9). Compound induced necrosis-like cell death (bright red fluorescence of ethidium bromide EB inside the damaged sessile cells in Fig. 9). Contrariwise, C. neoformans was resistant to 5h (arrows indicate weak green fluorescence of acridine orange AO inside the viable cells in Fig. 10).

Also the fact that the work is carried out with only 3 strains leaves some doubts. More strains should be used.

According to the  Reviewer’s comment, we will continue the studies on action mode of 5h with various fungal isolates. It will be presented in the next manuscript planned to be written and published in mdpi in near future.

Reference to these figures must be done in text. Readers need to understand what is intended to be shown with these pictures.

It was corrected.

Reviewer 2 Report

Staniszewska et al described a series of N-phenacyldibromobenzimidazoles and their mode of action as antifungal against C. albicans and C. neoformans

In my opinion,and although I believe this is an interesting and well-conceived work, there are some problems in this manuscript that should be addressed to for improve its quality and to reach the standard requisites of the journal. It is present form, this manuscript should be rejected, although resubmission could be considered.

- The aimed of the manuscript is to study the mode of action of a series of new synthesized N-phenacyldibromobenzimidazoles. Authors have carried out several biological assays to asses this aim, but only in a few of the new described compounds. Table 1 shows 20 compounds but initial antifungal activity data of 8 of them is shown in table 2. What about the others 12 compounds? Authors did not mention anywhere whether these derivatives have been tested or not. As claimed in reference 5, authors will report the Optimization of N-phenacyldibromobenzimidazole synthesis in another manuscript (in preparation), but biological data of all compounds should be described in this MS.

- Results section is quite confusing, specially section 2.8, where authors only give a little explanation of the results shown in figure 5, but nothing about figures 6-10.

- Legends in tables describing the biological data do not indicate how many experiments have been carried out to get the mean±SD.

- Legend in table 1 is also quite confusing, as I understand that superscript (a) only described compound 3c, and not the rest of them.

- Reference compound in all biological assays should be used.

- Discussion section should be also checked. Authors should clarify this sections as it is difficult to distinguished if they are describing the results or making hypothesis on them.

- I am not an expert in English grammar, but I have the impression authors extensively use the Saxon genitive to express the activity of compound 5h, and may be this is not correct. Authors should check this grammar issue.

Other errors:

- References format should be checked and reformatted.

 - For clarity and to facilitate the revison of the MS, I am attaching the original MS pdf where other comments and minor errors found have been highlighted

Author Response

Reviewer’s comments

Authors’ answers

- The aimed of the manuscript is to study the mode of action of a series of new synthesized N-phenacyldibromobenzimidazoles. Authors have carried out several biological assays to asses this aim, but only in a few of the new described compounds. Table 1 shows 20 compounds but initial antifungal activity data of 8 of them is shown in table 2. What about the others 12 compounds? Authors did not mention anywhere whether these derivatives have been tested or not. As claimed in reference 5, authors will report the Optimization of N-phenacyldibromobenzimidazole synthesis in another manuscript (in preparation), but biological data of all compounds should be described in this MS.

Biological data of all compounds presented in Table 2 and Fig. S1-S12 in Suppl., it was explained in the revised manuscript, please see below:

As it was shown in Table 2 and Fig. S1 -S12 (in Supplementary file), in our initial screening of twenty dibromobenzimidazole derivatives we assessed the percentage of cell growth inhibition (%I). The inhibitory concentration of 50% (IC50), the concentration of benzimidazoles that reduces the cell growth of C. albicans SC5314 by ≥50% was determined. Secondly, randomly selected (5f) and the most effective inhibitors (5e and 5h) were tested against the C. albicans SPZ176 isolate resistant to Flu and Itr (Table 2). 5e displayed IC50 at 4-16 µg/ml (Table 2) and the mode of fungicidal action against SC5314 at 8-16 µg/ml (lg R≤1.19 in Table 3). 5f showed lg R=1 at 8 µg/ml (Table 3). Contrariwise, 5h displayed no candidacidal action (lg R≤0.43 in Table 3). Moreover, a paradoxical growth phenomenon of the reference strain SC5314 [26] was noted for the following derivatives: 4f, 4h, 5a (Fig. S5, Fig. S7, Fig. S9, Fig. S11-S12) as well as 5b, 5e-f, 5h, 5j (Table 2). Briefly, we noted a slow decrease in the viable cell growth at higher concentrations (e.g., % I=53±8 at 16 µg/ml for 5b) vs the lowest concentrations at which the cell growth was substantially inhibited (e.g., % I=95±8 at 8 µg/ml for 5b).

- Results section is quite confusing, specially section 2.8, where authors only give a little explanation of the results shown in figure 5, but nothing about figures 6-10.

It was rewritten in the revised Ms, please see below:

The resulting cell wall damage and cell viability were assessed using Confocal laser scanning microscopy (CLSM) after treatment with 5h (twelve images were assessed for each treatment/ staining). As it was shown using CFW staining (Fig. 5), 5h at 16 µg/ml induced the cell wall rearrangement of the C. albicans sessile conglomerate. Biofilm’s chitin content was redistributed and elevated under treatment with 5h (vivid blue fluorescence of elevated chitin in Fig. 5). Contrariwise, action of 5h against the C. neoformans sessile growth was not significant (Fig. 6). In Fig. 6, very few cells were totally stained with CFW in conglomerate vs the untreated control showing several cells with bright blue fluorescence. Thus 5h did not reorganize the cell wall chitin content of C. neoformans.

Congo red (CR) interacts with β-D-glucan of the 5h-treated C. albicans sessile cells (Fig. 7). Thus the cells exposed to 5h at 16 µg/ml exhibit increased frequencies of the cell wall damage (arrows in Fig. 7). Contrariwise, the biofilm of C. neoformans treated with 5h was found CR sensitive in comparable level to the untreated sessile cells (Fig. 8). Thus 5h did not disturb glucan content of C. neoformans.

Compound 5h altered plasma membrane permeability which is indicated by intensive red fluorescence of the 5h-treated sessile cells (Fig. 9). Compound induced necrosis-like cell death (bright red fluorescence of ethidium bromide EB inside the damaged sessile cells in Fig. 9). Contrariwise, C. neoformans was resistant to 5h (arrows indicate weak green fluorescence of acridine orange AO inside the viable cells in Fig. 10).

- Legends in tables describing the biological data do not indicate how many experiments have been carried out to get the mean±SD.

Number of repetition was included in Table Legends and for CLSM analyses, we detailed as follows: The resulting cell wall damage and cell viability were assessed using Confocal laser scanning microscopy (CLSM) after treatment with 5h (twelve images were assessed for each treatment/ staining).

- Legend in table 1 is also quite confusing, as I understand that superscript (a) only described compound 3c, and not the rest of them.

The legend was removed and 2nd column in Table 1 was rewritten, see below:

3, Ar COCH2 X

3a, Ph, Br

3b, 4-FC6H4, Cl

3c, 4-ClC6H4, Br

3d, 4-BrC6H4, Cl

3e, 2,4-Cl2C6H3, Cl

3f, 3,4-Cl2C6H3, Cl

3g, 2,4,6-Cl3C6H2, Cl

3h, 2,4-F2C6H3, Cl

3i, 2,5-F2C6H3, Cl

3j, 2,4,6-F3C6H2, Cl

Additionally the Table 5 was rewritten.

- Reference compound in all biological assays should be used.

Simultaneously, results for AmB were presented in Table 2.

AmB results were included in Table 4

In Table 6, chitinase was used as ref.

Log cell growth reduction of C. albicans cells treated with AmB (stands for Amphotericin B used as a control antifungal agent), was described by our group previously: ‘AmB displayed MFC at concentrations ranging from 0.5 to 16 μg/ml.’ [Staniszewska M, Bondaryk M, Kazek M, Gliniewicz A, Braunsdorf C, Schaller M, Mora-Montes HM, Ochal Z. Effect of serine protease KEX2 on Candida albicans virulence under halogenated methyl sulfones. Future Microbiol. 2017 Mar;12:285-306. doi: 10.2217/fmb-2016-0141. Epub 2017 Feb 24. PMID: 28287299.] as well as the citation: [32] was included in the manuscript, please see explanation in the Discussion:

The AO/IP staining of cell treated with AmB was previously published by our group [Paulina Zielińska, Monika Staniszewska, Małgorzata Bondaryk, Mirosława Koronkiewicz, Zofia Urbańczyk-Lipkowska, Design and studies of multiple mechanism of anti-Candida activity of a new potent Trp-rich peptide dendrimers, European Journal of Medicinal Chemistry, Volume 105, 2015, Pages 106-119, https://doi.org/10.1016/j.ejmech.2015.10.013]. [Staniszewska Monika, Bondaryk Małgorzata, Wieczorek Magdalena, Estrada-Mata Eine, Mora-Montes Héctor M., Ochal Zbigniew.Antifungal Effect of Novel 2-Bromo-2-Chloro-2-(4-Chlorophenylsulfonyl)-1-Phenylethanone against Candida Strains. Frontiers in Microbiology.7:1309. 2016. DOI=10.3389/fmicb.2016.01309 https://www.frontiersin.org/article/10.3389/fmicb.2016.01309].

We studied the antifungal activity of dibromobenzimidazole without their enhancement by ref. compound. The action of AmB was extensively published previously and quoted in our manuscript, please see below:

In our study, the leading compound 5h (Table 4) was <16-times less active than AmB with minimal fungicidal concentration MFC90=1 µg/ml [32] and MFC=0.5 [33] against the C. albicans isolates and SC5314 respectively. Structure-activity relationships provide opportunities for synthesis of dibromobenzimidazole analogs with improved antifungal action. Moreover, the most active antifungals (5e-f, 5h) at the concentration range of 32-0.125 µg/ml were developed to generate viable and vital eukaryotic cells (Fig. 1 and Table S2- S3 and Fig. S1). Thus the tested dibromobenzimidazole were proved to be less cytotoxic against the Vero cells compared to AmB (toxic at 15-20 µg/ml after 24 h) [34].

In alignment with Górska-Nieć et al. [36], we proved that enhanced biomass production leads to loss of antifungal activity of 5h at concentrations ranging from 4 to 16 µg/ml. Moreover, the activity of 5h did not correspond with AmB affecting cell wall due to activity accompanied by an increase concentration in medium with sorbitol [37].

5h was able to hydrolase 4-nitrophenyl-N-acetyl-β-D-glucosaminide and 4-nitrophenyl-β-D-N,N’,N’’-triacetylchitothiose without activity against triacetylchitothiose (Table 6). Based on our results and in line with Nielsen and Sörensen [40], we hypothesized that 5h displays comparable activity to chitinase (EC 3.2.1.14). Since the ability of Congo red CR fluorescent tracker to visualize the fungal cell wall elements was described previously [41, 42], we used CR as a diazo compound pertaining to its high affinity to polysaccharides in the 5h-treated C. albicans [41, 42].

Furthermore, 5h induced the phosphatidylserine PE translocation and membrane permeability [42], these were shown using the Annexin V and propidium iodide PI staining assay (Fig. 3 and Fig. 4). We hypothesized that PE externalization affect subsequently the elevated chitin content (Fig. 5) and activity of 5h. Moreover, this polymer play an essential role in the sensitivity (or resistance) of C. albicans to AmB [43, 44].

We hypothesized that the adaptive response of C. neoformans showing elevated ROS production promotes stress resistance to 5h. Contrariwise, decrease in the ROS level by incubation with 5h can induce the lethal process adequately monitored by cytometric analysis (Fig 3-4 and Table S4). Based on the latter findings, 5h can act such as anti-oxidant. Moreover, elevated ROS under 5h correlated with fungicidal effect typical for AmB [45].

- Discussion section should be also checked. Authors should clarify this sections as it is difficult to distinguished if they are describing the results or making hypothesis on them.

Discussion was checked and rewritten: (1) original results were presented, (2) hypotheses were introduced, all changes marked in red, please see below:

In our study, the leading compound 5h (Table 4) was <16-times less active than AmB with minimal fungicidal concentration MFC90=1 µg/ml [32] and MFC=0.5 [33] against the C. albicans isolates and SC5314 respectively. Structure-activity relationships provide opportunities for synthesis of dibromobenzimidazole analogs with improved antifungal action. Moreover, the most active antifungals (5e-f, 5h) at the concentration range of 32-0.125 µg/ml were developed to generate viable and vital eukaryotic cells (Fig. 1 and Table S2- S3 and Fig. S1). Thus the tested dibromobenzimidazole were proved to be less cytotoxic against the Vero cells compared to AmB (toxic at 15-20 µg/ml after 24 h) [34].

In alignment with Górska-Nieć et al. [36], we proved that enhanced biomass production leads to loss of antifungal activity of 5h at concentrations ranging from 4 to 16 µg/ml. Moreover, the activity of 5h did not correspond with AmB affecting cell wall due to activity accompanied by an increase concentration in medium with sorbitol [37].

We showed that 5h displayed Candida spp dependent activity.

5h was able to hydrolase 4-nitrophenyl-N-acetyl-β-D-glucosaminide and 4-nitrophenyl-β-D-N,N’,N’’-triacetylchitothiose without activity against triacetylchitothiose (Table 6). Based on our results and in line with Nielsen and Sörensen [40], we hypothesized that 5h displays comparable activity to chitinase (EC 3.2.1.14). Since the ability of Congo red CR fluorescent tracker to visualize the fungal cell wall elements was described previously [41, 42], we used CR as a diazo compound pertaining to its high affinity to polysaccharides in the 5h-treated C. albicans [41, 42].

Furthermore, 5h induced the phosphatidylserine PE translocation and membrane permeability [42], these were shown using the Annexin V and propidium iodide PI staining assay (Fig. 3 and Fig. 4). We hypothesized that PE externalization affect subsequently the elevated chitin content (Fig. 5) and activity of 5h. Moreover, this polymer play an essential role in the sensitivity (or resistance) of C. albicans to AmB [43, 44].

We hypothesized that the adaptive response of C. neoformans showing elevated ROS production promotes stress resistance to 5h. Contrariwise, decrease in the ROS level by incubation with 5h can induce the lethal process adequately monitored by cytometric analysis (Fig 3-4 and Table S4). Based on the latter findings, 5h can act such as anti-oxidant. Moreover, elevated ROS under 5h correlated with fungicidal effect typical for AmB [45].

- I am not an expert in English grammar, but I have the impression authors extensively use the Saxon genitive to express the activity of compound 5h, and may be this is not correct. Authors should check this grammar issue.

English was corrected by native speaker and marked in red and highlighted in blue.

Other errors:

- References format should be checked and reformatted.

References were reformatted.

 - For clarity and to facilitate the revison of the MS, I am attaching the original MS pdf where other comments and minor errors found have been highlighted

Thank you for comments. These were included in the revised manuscript.

Round 2

Reviewer 1 Report

Table 3: there is a b) indicated in the legend, that is not seen in the table

There are still microorganisms’ denominations that are not in italic (see tables, legends and new text)

It is not still clear how a higher concentration result in a lower percentage of inhibition. Authors refer it on the text, but they do not give any explanation/hypothesis for this. They should do it

Authors say; “The most effective 5h at 16 µg/ml totally inhibited recovery of cfu of both clinical isolates” but that is not clear on table 4 as for Candida there are only one value of 105. It needs to be clarified.

The fact that the work is carried out with only 3 strains leaves some doubts. More strains should be used. Authors saying “we will continue the studies on action mode of 5h with various fungal isolates. It will be presented in the next manuscript planned to be written and published in mdpi in near future”, doesn´t bring any contribution to solve the doubts!! Having more results with different strains could help to explain the go up and down of percentage of inhibition with higher amount of compound. I strongly advise the use of more strains to robust and to give credit to this paper.

Reviewer 2 Report

Authors have clearly improved the quality of the MS and now reaches the standard requisites of the journal, so it can be considered for publication